# Unusual Trisomy X Phenotype Associated with a Concurrent Heterozygous 16p11.2 Deletion: Importance of an Integral Approach for Proper Diagnosis

**DOI:** 10.3390/ijms241914643

**Published:** 2023-09-27

**Authors:** Ariadna González-del Angel, Miguel Angel Alcántara-Ortigoza, Sandra Ramos, Carolina Algara-Ramírez, Marco Antonio Hernández-Hernández, Lorenza Saenger-Rivas

**Affiliations:** 1Laboratorio de Biología Molecular, Subdirección de Investigación Médica, Instituto Nacional de Pediatría, Secretaría de Salud, Mexico City CP 04530, Mexico; malcantaraortigoza@gmail.com; 2Facultad Mexicana de Medicina, Universidad la Salle, Mexico City CP 14070, Mexico; caroalgara@hotmail.com (C.A.-R.); marcots2099@gmail.com (M.A.H.-H.); lorenzasaenger@gmail.com (L.S.-R.); 3Laboratorio de Citogenética, Subdirección de Investigación Médica, Instituto Nacional de Pediatría, Secretaría de Salud, Mexico City CP 04530, Mexico; sera_ramos@yahoo.com.mx

**Keywords:** trisomy X, unusual phenotype, distal 16p11.2 chromosome deletion, copy-number variant, sex chromosome aneuploidy, autosomal recessive diseases

## Abstract

Trisomy X is the most frequent sex chromosome anomaly in women, but it is often underdiagnosed postnatally because most patients do not show any clinical manifestation. It is estimated that only 10% of patients with trisomy X are diagnosed by clinical findings. Thus, it has been proposed that the clinical spectrum is not yet fully delimited, and additional uncommon or atypical clinical manifestations could be related to this entity. The present report describes a female carrying trisomy X but presenting atypical manifestations, including severe intellectual disability, short stature, thymus hypoplasia, and congenital hypothyroidism (CH). These clinical findings were initially attributed to trisomy X. However, chromosome microarray analysis (CMA) subsequently revealed that the patient also bears a heterozygous 304-kb deletion at 16p11.2. This pathogenic copy-number variant (CNV) encompasses 13 genes, including *TUFM*. Some authors recommend that when a phenotype differs from that described for an identified microdeletion, the presence of pathogenic variants in the non-deleted allele should be considered to assess for an autosomal recessive disorder; thus, we used a panel of 697 genes to rule out a pathogenic variant in the non-deleted *TUFM* allele. We discuss the possible phenotypic modifications that might be related to an additional CNV in individuals with sex chromosome aneuploidy (SCA), as seen in our patient. The presence of karyotype-demonstrated trisomy X and CMA-identified 16p11.2 deletion highlights the importance of always correlating a patient’s clinical phenotype with the results of genetic studies. When the phenotype includes unusual manifestations and/or exhibits discrepancies with that described in the literature, as exemplified by our patient, a more extensive analysis should be undertaken to enable a correct diagnosis that will support proper management, genetic counseling, and medical follow-up.

## 1. Introduction

Trisomy X is the most frequent sex chromosome anomaly in women, affecting one in every thousand female newborns [1]. In most patients (58–63%), the extra X chromosome is due to a nondisjunction in maternal meiosis I; meanwhile, 16–17.4% of cases are due to an alteration in maternal meiosis II, and 18–19.6% of patients are due to postzygotic nondisjunction [2]. The most common karyotype is 47,XXX, but 10% of cases are reported to be mosaics consisting of two or more cell lines, including 46,XX/47,XXX, 47,XXX/48,XXXX, 45,X/47,XXX, and 45,X/46,XX/47,XXX [1,2].

Some trisomy X patients are diagnosed prenatally by karyotyping from amniocentesis or chorionic villus sampling [1], or through maternal serum-based fetal DNA analysis [2]. Such studies are usually requested for the detection of other chromosomal disorders, such as trisomy 21, 18, or 13. Postnatally, trisomy X is often underdiagnosed because most women carrying this alteration do not show any clinical manifestation [2].

Postnatal clinical diagnosis of trisomy X and its confirmation by peripheral blood cytogenetic study is usually performed when a patient presents neurodevelopmental delay, tall stature, and/or primary ovarian failure. However, it is estimated that only 10% of patients with trisomy X are diagnosed by clinical findings [1,2]. This suggests that the clinical spectrum is not yet fully delineated, and patients may have uncommon or atypical clinical manifestations [2].

The advent of chromosome microarray analysis (CMA) enabled the identification of an additional copy-number variant (CNV) that may explain unusual clinical findings in patients with aneuploidies or subtle rearrangements involving autosomes or sex chromosomes [3]. Here, we describe a female in whom trisomy X coexists with a CMA-identified 16p11.2 deletion that appears to explain some of the unusual clinical manifestations that were initially attributed to trisomy X. The applied diagnosis approach enabled us to provide more accurate genetic counseling, prognosis, and medical management.

## 2. Clinical Case

The patient is a Mexican 17 year old female who is the product of the first gestation of a 22 year old mother and a 28 year old father, both of whom were considered healthy and were not consanguineous. In the mother, preeclampsia was documented during the second trimester of gestation; this was treated with only a hyposodic diet. Oligohydramnios was identified at 32 weeks of gestation. The patient was born through cesarean at 34 weeks of gestation, with a weight of 2500 g (Z = −0.15), a height of 47 cm (Z = 0.09), and an APGAR score of 7/9.

Congenital hypothyroidism (CH) was identified through newborn screening. It was treated by hormone replacement therapy, which was initiated at 3 months of age when the patient began to show signs of respiratory distress. This respiratory distress required hospitalization for 3 months. During that time, the patient presented two cardiorespiratory arrests and began to have partial seizures.

The patient has been known to our service at the Instituto Nacional de Pediatría, México, since the age of 2 years. At the initial physical examination, we identified the following: head circumference, 45 cm (Z = −1.79); weight, 10.4 kg (Z = −1.55); height, 75 cm (Z = −3.22); forehead with hypertrichosis; bushy, arched eyebrows; depressed nasal bridge; bulbous nose (Figure 1a); hypoplastic helix in the left ear; short and broad neck; slight asymmetry of thorax at the expense of a more prominent right hemithorax; thoracolumbar scoliosis; aberrant bilateral palms; bilateral clinodactyly of the fifth finger.

At a medical follow-up, an echocardiogram revealed patent ductus arteriosus (PDA) of 3 × 9 × 6 mm, pulmonary arterial hypertension of 40 mmHg, and patent foramen ovale (PFO). Surgical management of PDA was performed at 2 years of age. An ultrasound performed at 2 years of age identified thymic hypoplasia (left lobule 1.7 × 1.4 mm, right lobe showing only isthmus) and thyroid hypoplasia (right lobe 11 × 8 mm and left lobe 6 × 5 mm), which agreed with the newborn-stage diagnosis of CH. Serum calcium and blood count were normal.

At the age of 2 years, a tracheostomy and gastrostomy were performed. At 3 years old, severe gastroesophageal reflux was treated by Nissen surgery. At 4 years old, auditory evoked potentials with a stimulus of 105 dB identified only the V wave and a hearing threshold of 105 dB; these findings were compatible with severe hearing loss. Cerebral and renal malformations were ruled out by computed tomography scan and ultrasound plus cystourethrography, respectively.

The patient required multiple hospitalizations due to respiratory infections up to the age of 9 years. Hypogammaglobulinemia was detected on one occasion when the patient was 2 years old (361 mg/dL, normal value 402–1008 mg/dL); it was treated with prophylactic intravenous gammaglobulin for 2 months. At 3 years of age, repeated pneumonias with normal IgG, IgA, and IgM levels and normal T lymphocyte, B lymphocyte, and NK cell counts (by flow cytometry) led to consider secondary immunodeficiency, probably due to her severe malnutrition.

At the time of this report, the patient is 17 years old (Figure 1b) and presents a head circumference of 49.5 cm (Z = −5.12), weight of 26.8 kg (Z = −9.36), and height of 127 cm (Z = −5.55). The patient has no head support, sitting ability, or sphincter control; she has a social smile, fixes her gaze, partially follows objects with her eyes, takes objects with one hand, and has been saying monosyllables since the age of 5 years. She has been a custodial patient since she was 3 years old; she has spastic quadriparesis and undergoes seizure control with administration of levetiracetam and lamotrigine. She presented thelarche and pubarche at 10 years old, and menarche at 10 years and 10 months old. Currently, the patient has regular menstrual cycles (every 30 days with a length of 5 days), with Tanner breast and pubic scores of V and IV, respectively.

## 3. Genetic Analysis

Due to the presence of delayed neurodevelopment with seizures, severe hearing loss, short stature, facial dysmorphism, and alterations in the thymus, thyroid, and heart, a diagnostic approach was initiated at 2 years of age. Karyotyping of G-banded peripheral blood lymphocytes showed trisomy X: 47,XXX in 15 metaphases with a resolution of 450 bands (Figure 2a,b).

As thymus hypoplasia and CH are not clinical manifestations described in trisomy X, other genetic entities were assessed. Fluorescence in situ hybridization (FISH) analysis was used to rule out velocardiofacial syndrome, as normal ish 22q11.2(TUPLE 1x2) was found in 25 metaphases (Figure 2c). Complete Sanger automated sequencing did not identify any pathogenic variant in genes associated with other hypothyroidism-relevant syndromic entities (i.e., *PAX8* and *FOXE1*) (images of electropherograms are available upon request). CMA (ClariSure^®^ Oligo-SNP, 2.67 million probes, Test code: 16478; Quest Diagnostics, Nichols Institute, Chantilly, VA, USA) confirmed trisomy X but also identified a heterozygous 304-kb deletion at 16p11.2 (arr[GRCh37] (X)x3,16p11.2(28,747,521_29,051,191)x1) (Figure 3a). A Leukodystrophy and Genetic Leukoencephaly Panel of 697 genes (Invitae, Test code: 55002) did not identify any pathogenic variant in the non-deleted *TUFM* allele of the patient.

Karyotyping of G-banded peripheral blood lymphocytes was normal in both parents at a resolution of 400–550 bands. In order to determine the origin of the 16p11.2 deletion, four microsatellite (STR) loci linked to *SH2B1* (https://www.ncbi.nlm.nih.gov/dbvar/ accessed on 24 August 2023: nsv1826241 and nsv1826246), *ATP2A1* (nsv1826454), and *CD19* (nsv1826465) genes, which are involved in the 16p11.2 deletion’s interval, were analyzed by polymerase chain reaction (PCR) and capillary electrophoresis (employed primer sequences and PCR conditions are available upon request). The maternal and paternal heterozygous genotype documented at nsv1826454 (*ATP2A1*), and nsv1826246 (*SH2B1*) STR loci, supported a de novo origin of the 16p.11.2 deletion (Figure 3b), although its parental origin could not be determined.

## 4. Discussion

Despite the high frequency of trisomy X (1:1000 female newborns), it is usually underdiagnosed since, unlike in other chromosomopathies, most of these patients present without any clinical manifestation [1]. Postnatal diagnosis of trisomy X by karyotype is typically prompted by the presence of neurodevelopmental delay (63%), language disorder (13%), behavioral disturbance (7%), hypotonia (3.5%), and/or facial dysmorphism (3.5%) [2]. The most common dysmorphisms include hypertelorism, internal epicanthus, clinodactyly of the fifth finger, and pes planus/flat feet [1]. From among these clinical manifestations, our patient presented only a severe intellectual disability and clinodactyly of the fifth finger. This could reflect the variable expressivity of trisomy X in which most of the patients do not present an abnormal phenotype [1].

Some studies of patients with trisomy X found that their intelligence quotient (IQ) can range from 55 to 115, and 15% of patients experience absence, partial, or generalized seizures [1]. In this light, our patient presented uncharacteristically severe intellectual disability. Prior to obtaining the CMA results, we considered that this could reflect her history of multiple respiratory infections requiring hospitalization during childhood and/or her cardiopulmonary arrests during the first 6 months of life.

Tall stature is another reason for consult among patients with trisomy X (≥centile 75, 80–89%) [1]. Notably, our patient’s height was below centile 3. However, this could be related to her history of gastrostomy-tube feeding from the first months after birth.

Up to 23% of patients with trisomy X present a congenital malformation, such as cleft palate, velopharyngeal insufficiency, cardiopathy, and/or genitourinary anomaly [1]. In our patient, the presence of PDA and PFO could be due to the trisomy X, given that such patients have been described as having congenital cardiopathies, including auricular or ventricular septal defects, pulmonary stenosis, and coarctation of the aorta. However, the literature indicates that the frequency of cardiopathies in trisomy X is not higher than that observed in the general population (8–9 per 1000 newborns) [2,4].

Thymic hypoplasia has not been previously described in trisomy X, and, as far as we know, CH has been observed in only a single Brazilian female patient with trisomy X, who also presented short stature, intellectual disability, hypodontia, tooth fusion, and bladder diverticulum with vesicoureteral reflux [5]. The presence of trisomy X and CH may be coincidental, especially in our patient, given the high birth prevalence reported for this entity in the Mexican population (1:1373) [6].

Patients with trisomy X present adequate pubertal development during adolescence with normal fertility, but exhibit a high frequency of premature ovarian insufficiency (1 per 900 patients) compared to the general population [7]. It is estimated that trisomy X is the causative factor in 3.8% of patients with premature ovarian failure, making it one of the most common genetic causes of infertility. Our patient exhibited normal pubertal development according to previous descriptions [8,9], and she has regular menstrual cycles.

The development of CMA has enabled apparently atypical phenotypes of some aneuploidies to be explained through the identification of an additional CNV that was not identified by conventional karyotyping [3]. Given that our patient had findings that would be considered uncommon for trisomy X, such as severe intellectual disability, hearing loss, short stature, thymus hypoplasia, and CH, we performed a CMA analysis. This led to the identification of a cytogenetically indistinguishable deletion of 304 kb involving 16p11.2 (Figure 3a), which was classified as a pathogenic CNV by the ClinGen CNV Interpretation Calculator online tool (total score: 1.75; https://cnvcalc.clinicalgenome.org/cnvcalc/, accessed on 24 August 2023).

Two microdeletions at 16p11.2 have been extensively described in the literature: a proximal microdeletion involving 593 kb (29.5 Mb to 30.1 Mb) and a smaller distal microdeletion of 220 kb (28.74 Mb to 28.95 Mb) [10,11,12]. The proximal microdeletion has a reported prevalence of 0.5% among patients with neurological alterations. Individuals with this deletion present autism (16.1–25.6%), intellectual disability (10.3–28.1%), psychomotor delay (50–57%), and seizure (19.4–29%). These patients typically have facial dysmorphism and congenital malformations (21.1–58.5%), including macrocephaly, increased subarachnoid space, and tonsillar cerebellar ectopia (Chiari I), as well as sacral dimples (34%), brown spots (30%), and obesity [10,11]. The distal 16p11.2 microdeletion has a prevalence of 0.3% in patients with intellectual disability. It is associated mainly with delayed neurodevelopment (in a third of cases), behavioral disturbance (16%), seizure (9%), facial dysmorphism (22.7%, including prominent forehead, downward palpebral fissures, narrow palpebral opening), and obesity (18%) [12].

The 16p11.2 de novo microdeletion of 304 kb observed in our patient is considered distal, but it is larger than the previously described distal microdeletion (28.74 to 29.05 Mb vs. 28.74 to 28.95 Mb) [12]. Some of the clinical manifestations in our patient that appear to be attributable to the 16p11.2 microdeletion include the neurological disability and seizures. She did not share the behavioral disturbance, facial dysmorphism, or obesity described in this entity (Table 1), which may reflect the clinical variability associated with the distal 16p11.2 microdeletion. We also cannot rule out that the coexistence of two genetic alterations (a “two-hit” hypothesis) could modify the phenotypes expected on the basis of their independent presentations [3].

The severe neurological affection and the atypical phenotype seen in our patient, including the presence of facial dysmorphism that is not described in trisomy X or in the 16p11.2 microdeletion (Table 1), could be consistent with those observed by Le Gall et al. [3]. These authors described 14 patients with sex chromosome aneuploidy (SCA), including four with trisomy X, having a severe or atypical phenotype consisting of intellectual disability, hypotonia, seizures, facial dysmorphism, and/or congenital malformation associated with an additional CNV. The authors suggested that the presence of a CNV, especially one classified as pathogenic or likely pathogenic, might be considered as an additional genetic factor contributing to the phenotypic variability among patients with SCA, particularly in those exhibiting more severe intellectual disability. Thus, the authors proposed that patients with SCA should be investigated for the coexistence of other genetic events, such as CNVs, which were found at a frequency of 14.8% in another group of 27 SCA individuals [3].

However, Mountford et al. subsequently failed to replicate the observations reported by Le Gall et al. The former group found nine pathogenic CNVs in patients with sex chromosome trisomies (4/43 XYY, 4/42 XXY, and 1/40 XXX) but did not identify more severe intellectual disability in their population. The authors suggested that the discrepancy with the finding of Le Gall et al. might be related to differences in the utilized selection criteria and to the well-known highly variable phenotype, which is conditioned by several pathogenic CNVs that even includes asymptomatic carriers. Furthermore, the authors did not discard the possibility that the coexistence of specific CNVs might modify the phenotype of only certain genetic abnormalities [13]. In addition to our patient, the literature contains reports of six trisomy X patients carrying an additional CNV. None of them share the same CNV, making it difficult to suggest any clear phenotype/genotype correlation [3,13,14].

The 16p11.2 microdeletion described herein encompasses 13 genes (https://www.genedistiller.org/ accessed on 24 August 2023, Figure 3a). Among them, *TUFM*, *SH2B1*, *CD19*, and *ATP2A1* draw attention considering the phenotypes associated with their pathogenic genetic variants.

*TUFM* encodes the mitochondrial translation elongation factor, EF-Tu, which is a GTPase. To date, only four patients have been reported with pathogenic variants in this gene; they exhibit deficiency of oxidative phosphorylation with an autosomal recessive inheritance pattern (COXPD4, OMIM #610678). The most important clinical manifestations are early- and severe-onset lactic acidosis, progressive lethal encephalopathy, and cardiomyopathy [15]. Some authors recommend that, when a phenotype differs from that described for an identified microdeletion, the presence of pathogenic variants in the non-deleted allele should be ruled out to assess for an autosomal recessive disorder [16]. Although our patient did not have clinical manifestations compatible with COXPD4, we requested the application of a genetic panel that includes the *TUFM* gene, in order to assess this as a possible cause of her severe intellectual disability. This analysis did not identify a pathogenic variant in the non-deleted *TUFM* allele.

The *SH2B1* gene (OMIM*608937) plays an important role in the development of obesity. It encodes a protein involved in the insulin and leptin signaling pathways, and patients with proximal or distal deletions in 16p11.2 tend to develop obesity at an early age [12]. This feature was absent in our patient, despite her physical disability. However, given that obesity is a multifactorial trait that can be modified by environmental factors, such as feeding, the lack of obesity in our patient might reflect that she has been fed by nasogastric tube since the first months of life.

Pathogenic variants in *CD19* are responsible for common variable immunodeficiency type 3, which shows autosomal recessive inheritance (OMIM #613493) and conditions a severe hypogammaglobulinemia that causes recurrent bacterial infections [17]. Our patient presented aspiration pneumonias due to a gastroesophageal reflux that required Nissen surgery and tube feeding, as well as bacterial pneumonias in infancy. However, there was no identified persistent alteration of immunoglobulins, lymphocytes, or NK cells in blood, ruling out a primary immunodeficiency diagnosis.

Autosomal recessive myopathy, known as Brody’s disease (OMIM #601003), is due to pathogenic variants in *ATP2A1*, which encodes a calcium-dependent ATPase found in the sarcoplasmic reticulum of skeletal muscle (SERCA 1). Its clinical diagnosis is suggested by the presence of exercise-induced muscle stiffness in limbs and eyelids, onset in childhood, normal or slightly elevated levels of creatine phosphokinase (CPK), absence of myotonic discharge on electromyography, and mild myopathic changes with atrophy of type II fibers on muscle biopsy; in addition, patients are at risk of developing malignant hyperthermia [18]. Our patient lacked any reported malignant hyperthermia or episode of stiffness or delayed muscle relaxation after physiotherapy; thus, this diagnosis was not clinically supported.

Other clinical findings, such as the severe hearing loss, short stature, thymus hypoplasia, and CH, do not appear to be explained by either of the two genetic abnormalities. We consider that the patient’s severe hearing loss and short stature may reflect her history of multiple adverse events. The thymus hypoplasia and CH might be coincidental findings, given that we discarded a microdeletion in 22q11 as the cause of the thymus hypoplasia, and CH due to thyroid dysgenesis is a common entity in Mexico. Pathogenic variants in *PAX8* and *FOXE1* have been documented in a very low proportion of Mexican patients with CH (2.5%) [19], but we discarded this possibility by complete Sanger sequencing of the loci. It remains possible that other single-gene variations may synergistically contribute to the clinical manifestations in our patient, as recently exemplified by the association of *TIMP3* risk variants with bicuspid aortic valve development in patients with Turner syndrome [20].

## 5. Conclusions

The presence of karyotype-documented trisomy X and CMA-identified 16p11.2 deletion highlights that a patient’s clinical phenotype should always be correlated with the results of genetic studies. When there are uncommon manifestations and/or discrepancies between the patient’s phenotype and that described in the literature, as exemplified by our patient, a more extensive genetic approach should be considered, with the goal of providing a correct diagnosis that allows for accurate management, genetic counseling, and medical follow-up.

## Figures and Tables

**Figure 1 ijms-24-14643-f001:**
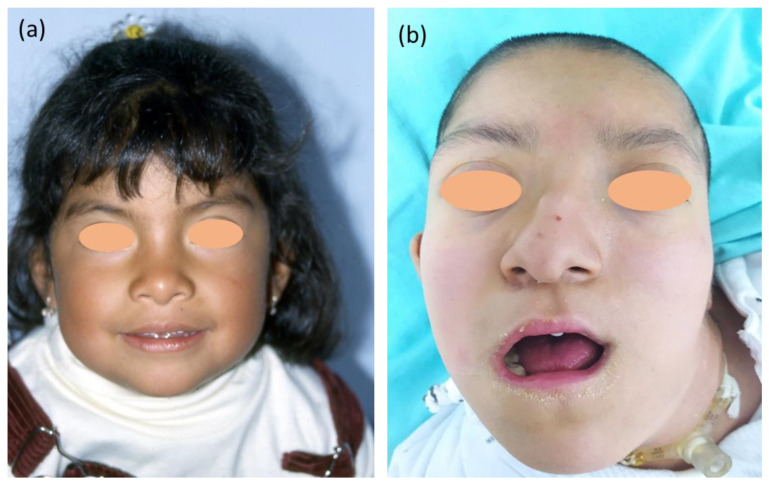
Clinical phenotype of the patient. (**a**) The patient at 2 years of age, presenting minor dysmorphias including bushy, arched eyebrows (HP:0000574), depressed nasal bridge (HP:0005280), and bulbous nose (HP:0000414). (**b**) The patient at 17 years of age, showing a long face (HP:0000276), bushy and arched eyebrows (HP:0000574), and a prominent nasal septum (HP:0005322).

**Figure 2 ijms-24-14643-f002:**
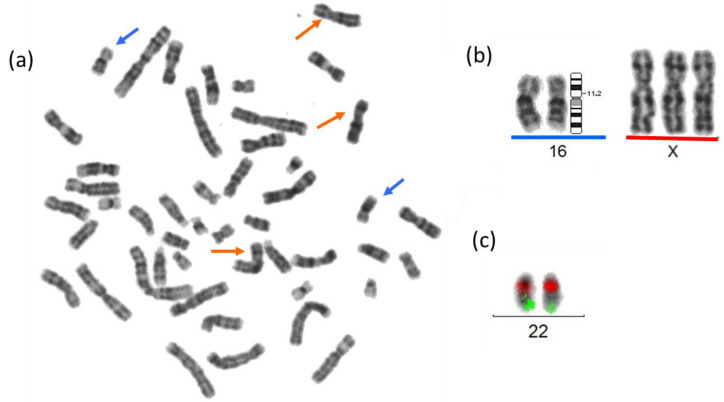
Cytogenetic analysis of the present patient. (**a**) Peripheral blood metaphase: 47,XXX (the X chromosomes are indicated with orange arrows and the chromosomes 16 with blue arrows). (**b**) Partial karyotype, trisomy X is shown above the red line, the pair of chromosomes 16 is shown above the blue line; the region 16p11.2 involved in the deletion identified by CMA is indicated in the ideogram. (**c**) A partial karyotype after FISH with a dual TUPLE1 spectrum orange/ARSA spectrum green probe (Vysis^®^) to discard a deletion in 22q11.2. Both normal 22 chromosomes have red and green signals.

**Figure 3 ijms-24-14643-f003:**
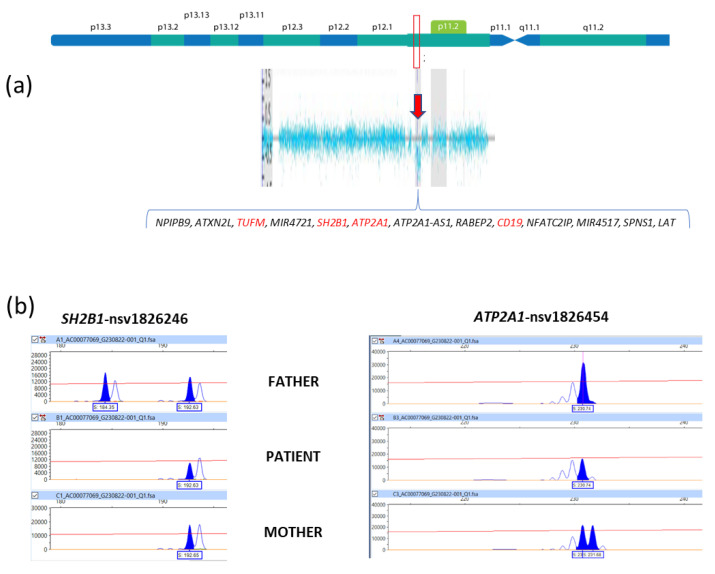
Molecular analysis of the present patient. (**a**) CMA: the arrow shows the heterozygous 304-kb deletion at 16p11.2 and the genes involved in the deletion are indicated, in red letters are the ones mentioned in the discussion. (**b**) Electropherograms of the two informative STR markers that allowed determining a de novo origin of the 16p11.2 deletion, by means of heterozygous genotypes identified in the patient´s father (nsv1826246) and mother (nsv1826454). The assigned alleles are highlighted in blue.

**Table 1 ijms-24-14643-t001:** List comparing clinical manifestations present in trisomy X, 16p11 distal deletion, and our patient, using codes of Human Phenotype Ontology (HP).

Clinical Manifestation	Trisomy X	16p11.2 Distal Deletion	Present Patient
**Neurodevelopmental delay HP:0012758/intellectual disability HP:0001249**	+	+	+
**Autistic behavior HP:0000729**	-	+	-
**Behavioral abnormality HP:0000708**	+	+	-
**Epilepsy HP:0001250**	+	+	+
**Hypotonia HP:0001252**	+	+	-
**Tall stature HP:0000098**	+	-	-
**Obesity tendency HP:0001513**	-	+	-
**Minor dysmorphias**	-	+	-
Prominent forehead HP:0011220	+	-	-
Upwards palpebral fissures HP:0000582	-	+	-
Down-slanted palpebral fissures HP:0000494	-	+	-
Narrow palpebral fissures HP:0045025	+	-	-
Internal epicanthus HP:0000286	+	-	-
Depressed nasal bridge HP:0005280	-	-	+
Bulbous nose HP:0000414	-	-	+
Short neck HP:0000470	-	-	+
Broad neck HP:0000475	-	-	+
Asymmetry on thorax HP:0000765	-	-	+
Thoracolumbar scoliosis HP:0002944	-	-	+
Abnormality of the palmar creases HP:0010490	-	-	+
Clinodactyly of the 5th finger HP:0004209	+	-	+
Pes planus HP:0001763	+	-	-
**Major dysmorphias (malformations)**			
Cerebral HP:0002060	+	+	-
Cardiac HP:0001627	+	-	+
Renal HP:0000077	+	+	-
**Other alterations:**			
Primary ovarian failure HP:0008209	+	-	NA
Thymic hypoplasia HP:0000778	-	-	+
Congenital hypothyroidism HP:0000851	- *	-	+

Human Phenotype Ontology (HP) [HPO: code of Human Phenotype Ontology (https://hpo.jax.org/app/ accessed on 24 August 2023)]. Present, +; absent, -; not applicable, NA. * Described in only one Brazilian patient [5].

## Data Availability

The datasets used and/or analyzed during this study are available from the corresponding author upon request.

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
