# Peer review of "Unusual Trisomy X Phenotype Associated with a Concurrent Heterozygous 16p11.2 Deletion: Importance of an Integral Approach for Proper Diagnosis"

_ijms, 2023, doi:10.3390/ijms241914643_

Round 1

Reviewer 1 Report

A case of an unusual phenotype in a patient with a 47,XXX karyotype due to the presence of a heterozygous deletion 16p11.2 in the genome is presented. The authors provide a detailed dynamic clinical description of the patient and present a comparative table of the clinical manifestations of the patient and those with isolated trisomy X and 16p11.2 distal deletion. In the discussion, the authors consider the features of the phenotype caused by the presence of additional copy of the X chromosome in the patient's genome and the haploinsufficiency of individual genes that are part of the 16q11.2 distal deletion. In conclusion, the authors emphasize the importance of conducting an extended genetic study if the patient exhibits a clinical phenotype that does not correspond to the identified chromosomal pathology.

General comment:

The manuscript possesses undeniable scientific significance; nevertheless, the "discussion" and "conclusions" sections in this particular version lack sufficient informativeness.

It is known that 16p11.2 distal deletion is inherited from phenotypically normal parents in more than 50% of cases. Therefore, it is crucial to determine its origin in the patient, as this can significantly alter the focus of the discussion in this study. If the given CNV is inherited from one of the parents, it may help explain why a large duplication of the entire X chromosome can have widely variable effects (following a 'two-hit' model for developmental delay).

The authors have conducted a comprehensive comparative analysis of clinical manifestations in the patient using Human Phenotype Ontology data. While Table 1 contains interesting information, it is discussed rather briefly. It would be valuable to discuss and explain the presence of certain facial abnormalities in the patient that are absent in isolated trisomy X and del16p11.2 сases, as well as the absence of certain features in the patient that are observed in isolated trisomy X and del16p11.2 (such as behavioral abnormality and malformations).

Addressing the questions "Are these CNVs related?" and "Do the two CNVs interact with each other or are they simply additive?" would undoubtedly add more scientific novelty to this manuscript than a simple and well-known statement about the need for a more extensive genetic approach.

Minor Remarks:

          page 2, line 47: it is advisable to replace the term "chromosomal formula" with the term "karyotype."

          It is recommended to classify the detected del16p11.2 into classes based on the degree of likelihood of pathogenicity (e.g., uncertain significance, likely pathogenic, pathogenic).

Author Response

August 24th 2023.

Reviewer 1:

A case of an unusual phenotype in a patient with a 47,XXX karyotype due to the presence of a heterozygous deletion 16p11.2 in the genome is presented. The authors provide a detailed dynamic clinical description of the patient and present a comparative table of the clinical manifestations of the patient and those with isolated trisomy X and 16p11.2 distal deletion. In the discussion, the authors consider the features of the phenotype caused by the presence of additional copy of the X chromosome in the patient's genome and the haploinsufficiency of individual genes that are part of the 16q11.2 distal deletion. In conclusion, the authors emphasize the importance of conducting an extended genetic study if the patient exhibits a clinical phenotype that does not correspond to the identified chromosomal pathology.

General comment:

The manuscript possesses undeniable scientific significance; nevertheless, the "discussion" and "conclusions" sections in this particular version lack sufficient informativeness.

RESPONSE: We thank the reviewer for these valuable suggestions. Our point-by-point responses are shown below.

It is known that 16p11.2 distal deletion is inherited from phenotypically normal parents in more than 50% of cases. Therefore, it is crucial to determine its origin in the patient, as this can significantly alter the focus of the discussion in this study. If the given CNV is inherited from one of the parents, it may help explain why a large duplication of the entire X chromosome can have widely variable effects (following a 'two-hit' model for developmental delay).

Response: We determined that the 16p11.2 deletion was de novo, the images of the molecular analysis was added in figure 3b.

Karyotyping of G-banded peripheral blood lymphocytes was normal in both parents at a resolution of 400-550 bands. In order to determine the origin of the 16p11.2 deletion, four microsatellite (STR) loci linked to SH2B1 (https://www.ncbi.nlm.nih.gov/dbvar/: nsv1826241 and nsv1826246), ATP2A1 (nsv1826454), and CD19 (nsv1826465) genes, which are involved in the 16p11.2 deletion´s interval, were analyzed by polymerase chain reaction (PCR) and capillary electrophoresis (employed primer sequences and PCR conditions are available upon request). The maternal and paternal heterozygous genotype documented at nsv1826454 (ATP2A1), and nsv1826246 (SH2B1) STR loci, respectively, supported a de novo origin of the 16p.11.2 deletion (Figure 3b), although its parental origin could not be determined.

The authors have conducted a comprehensive comparative analysis of clinical manifestations in the patient using Human Phenotype Ontology data. While Table 1 contains interesting information, it is discussed rather briefly. It would be valuable to discuss and explain the presence of certain facial abnormalities in the patient that are absent in isolated trisomy X and del16p11.2 сases, as well as the absence of certain features in the patient that are observed in isolated trisomy X and del16p11.2 (such as behavioral abnormality and malformations).

Addressing the questions "Are these CNVs related?" and "Do the two CNVs interact with each other or are they simply additive?" would undoubtedly add more scientific novelty to this manuscript than a simple and well-known statement about the need for a more extensive genetic approach.

Response: We consider that it is difficult to have a clear explanation about the modification of the phenotype in our patient, although, according to literature, we added in the discussion section some possible explanations about the phenotype modification in our patient in relation to the phenotype previously described in Trisomy X and del 16p11.2 and we also discussed some data about the presence of two genetic alterations in an individual and their possible interactions ”two hit effect” as follows:

The 16p11.2 de novo microdeletion of 304 kb observed in our patient is considered distal, but it is larger than the previously described distal microdeletion (28.74 to 29.05 Mb vs. 28.74 to 28.95 Mb) [12]. Some of the clinical manifestations in our patient that appear to be attributable to the 16p11.2 microdeletion include the neurological disability and seizures. She did not share the behavioral disturbance, facial dysmorphism or obesity de-scribed in this entity (Table 1), which may reflect the clinical variability associated with the distal 16p11.2 microdeletion. We also cannot rule out that the coexistence of two genetic alterations (a “two-hit” hypothesis) could modify the phenotypes expected based on their independent presentations [3].

The severe neurological affection and the atypical phenotype seen in our patient, including the presence of facial dysmorphism that is not described in trisomy X or in the 16p11.2 microdeletion (Table 1), could be consistent with those observed by Le Gall et al. in 2017 [3]. These authors described 14 patients with sex chromosome aneuploidy (SCA), including four with trisomy X, having a severe or atypical phenotype consisting of intellectual disability, hypotonia, seizures, facial dysmorphism, and/or congenital malformation associated with an additional CNV. The authors suggested that the presence of a CNV, especially those classified as pathogenic or likely pathogenic, might be considered as an additional genetic factor contributing to the phenotypic variability among patients with SCA, particularly in those exhibiting more severe intellectual disability. Thus, the authors proposed that patients with SCA should be investigated for the coexistence of other genetic events, such as CNVs, which were found at a frequency of 14.8% in another group of 27 SCA individuals [3].

However, Mountford et al. subsequently failed to replicate the observations reported by Le Gall et al. The former group found nine pathogenic CNVs in patients with sex chromosome trisomies (4/43 XYY, 4/42 XXY, and 1/40 XXX) but did not identify more severe intellectual disability in their population. The authors suggested that the discrepancy with the finding of Le Gall et al. might be related to differences in the utilized selection criteria and to the well-known highly variable phenotype, which is conditioned by several pathogenic CNVs that even includes asymptomatic carriers. Also, the authors did not discard the possibility that the coexistence of specific CNVs might modify the phenotype of only certain genetic abnormalities [13]. In addition to our patient, the literature contains reports of six trisomy X patients carrying an additional CNV. None of them share the same CNV, making it difficult to suggest any clear phenotype/genotype correlation [3,13,14].

Other clinical findings, such as the severe hearing loss, short stature, thymus hypoplasia, and CH, do not appear to be explained by neither of the two genetic abnormalities. We consider that the patient’s severe hearing loss and short stature may reflect her history of multiple adverse events. The thymus hypoplasia and CH might be coincidental findings, given that we discarded a microdeletion in 22q11 as the cause of the thymus hypoplasia, and CH due to thyroid dysgenesis is a common entity in Mexico. Pathogenic variants in PAX8 and FOXE1 have been documented in a very low proportion of Mexican patients with CH (2.5%) [19], but we discarded this possibility by complete Sanger sequencing of the loci. It remains possible that other single-gene variations may synergistically contribute to the clinical manifestations in our patient, as recently exemplified by the as-sociation of TIMP3 risk variants with bicuspid aortic valve development in patients with Turner syndrome [20].

Minor Remarks:

Reviewer: •          page 2, line 47: it is advisable to replace the term "chromosomal formula" with the term "karyotype."

Response: Thanks for your suggestion, we changed “chromosomal formula” for “karyotype”

  • It is recommended to classify the detected del16p11.2 into classes based on the degree of likelihood of pathogenicity (e.g., uncertain significance, likely pathogenic, pathogenic).

Response: The 16p11.2 microdeletion of 304 kb observed in our patient was classified as a pathogenic CNV by the ClinGen CNV Interpretation Calculator online tool. This classification was mentioned in the discussion section as follow:

This led to the identification of a cytogenetically indistinguishable deletion of 304 kb in-volving 16p11.2 (Figure 3a), which was classified as a pathogenic CNV by the ClinGen CNV Interpretation Calculator online tool (total score: 1.75; https://cnvcalc.clinicalgenome.org/cnvcalc/, accessed August 24, 2023).

Reviewer 2 Report

The authors report a case of a female patient with multiple clinical conditions, including intellectual disability, thymic hypoplasia, congenital hypothyroidism, and cardiac dysmorohism. The patient was initially diagnosed with congenital hypothyroidism via newborn screening. Later at the age of 2 years karyotyping of peripheral blood lymphocytes revealed trisomy X: 47,XXX. However, the observed clinical manifestations were atypical for this chromosome syndrome. Further investigation through chromosome microarray analysis (CMA) identified an additional heterozygous 304-kb deletion at 16p11.2 in the patient, which partially explained some of the clinical findings.

The case report is complete and clearly written; there are few minor issues that need clarification.

1.     To enhance the manuscript, it would be beneficial to discuss whether chromosome aneuploidies, particularly trisomy X, had been reported to be accompanied by other chromosomal rearrangements, such as microdeletions.

2.     Figure 1d should be updated to include chromosome coordinates or a relevant scheme to clarify the association  of the shown deletion with 16p11.2 region. Additionally, incorporating inserts for chromosome 16 in Figure 1c would illustrate the limitations of  G-banded karyotyping.

3.     It is a good practice to include all referred data, even if these analyses did not reveal any abnormalities. Karyotyping of peripheral blood lymphocytes in both parents,  FISH for 22q11.2 abnormalities in the patient and other mentioned analyses, provided as supplementary data, would enhance the completeness and transparency of the study. 

Author Response

Reviewer2:

The authors report a case of a female patient with multiple clinical conditions, including intellectual disability, thymic hypoplasia, congenital hypothyroidism, and cardiac dysmorohism. The patient was initially diagnosed with congenital hypothyroidism via newborn screening. Later at the age of 2 years karyotyping of peripheral blood lymphocytes revealed trisomy X: 47,XXX. However, the observed clinical manifestations were atypical for this chromosome syndrome. Further investigation through chromosome microarray analysis (CMA) identified an additional heterozygous 304-kb deletion at 16p11.2 in the patient, which partially explained some of the clinical findings.

The case report is complete and clearly written; there are few minor issues that need clarification.

Response: We thank the reviewer for these valuable suggestions. Our point-by-point responses are shown below.

Reviewer.     To enhance the manuscript, it would be beneficial to discuss whether chromosome aneuploidies, particularly trisomy X, had been reported to be accompanied by other chromosomal rearrangements, such as microdeletions.

Response: According to our revision of literature, beside the present case, there are only 6 patients with trisomy X that also have other microdeletions identified by chromosomal microarray analysis (CMA), but they do not share the same variable number copy (CNV). We mention this information in the discussion section as follows:

The severe neurological affection and the atypical phenotype seen in our patient, including the presence of facial dysmorphism that is not described in trisomy X or in the 16p11.2 microdeletion (Table 1), could be consistent with those observed by Le Gall et al. in 2017 [3]. These authors described 14 patients with sex chromosome aneuploidy (SCA), including four with trisomy X, having a severe or atypical phenotype consisting of intellectual disability, hypotonia, seizures, facial dysmorphism, and/or congenital malformation associated with an additional CNV. The authors suggested that the presence of a CNV, especially those classified as pathogenic or likely pathogenic, might be considered as an additional genetic factor contributing to the phenotypic variability among patients with SCA, particularly in those exhibiting more severe intellectual disability. Thus, the authors proposed that patients with SCA should be investigated for the coexistence of other genetic events, such as CNVs, which were found at a frequency of 14.8% in another group of 27 SCA individuals [3].

However, Mountford et al. subsequently failed to replicate the observations reported by Le Gall et al. The former group found nine pathogenic CNVs in patients with sex chromosome trisomies (4/43 XYY, 4/42 XXY, and 1/40 XXX) but did not identify more severe intellectual disability in their population. The authors suggested that the discrepancy with the finding of Le Gall et al. might be related to differences in the utilized selection criteria and to the well-known highly variable phenotype, which is conditioned by several pathogenic CNVs that even includes asymptomatic carriers. Also, the authors did not discard the possibility that the coexistence of specific CNVs might modify the phenotype of only certain genetic abnormalities [13]. In addition to our patient, the literature contains reports of six trisomy X patients carrying an additional CNV. None of them share the same CNV, making it difficult to suggest any clear phenotype/genotype correlation [3,13,14].

  1. Figure 1d should be updated to include chromosome coordinates or a relevant scheme to clarify the association of the shown deletion with 16p11.2 region. Additionally, incorporating inserts for chromosome 16 in Figure 1c would illustrate the limitations of G-banded karyotyping.

Response: We added figures 2 and 3 to clarify the 16p11.2 region involved in the deletion We added in figure 2 an ideogram of chromosome 16 in order to illustrate that the deletion is not visible by this technique. We also added in figure 3 a scheme to show the 16p11.2 region deleted in our patient that was identified by the chromosomal microarray analysis.

  1. It is a good practice to include all referred data, even if these analyses did not reveal any abnormalities. Karyotyping of peripheral blood lymphocytes in both parents, FISH for 22q11.2 abnormalities in the patient and other mentioned analyses, provided as supplementary data, would enhance the completeness and transparency of the study.

Response: We agree with your comment, according with that, we added in figure 2 an image of FISH for 22q112. We mentioned in the genetic analysis section that the files of the normal electropherograms of Sanger sequencing of PAX8 and FOXE1 genes are available upon request.

Unfortunately, the karyotypes of the parents were done some years ago and we do not have images of them, we have only the files with the written reports that mention they were normal. We contact the parents to do the analysis to establish that the deletion 16p11.2 was de novo (Figure 3b) as suggested by other reviewer, but they only accepted the sampling of the oral mucosa, so we do not have blood samples of both parents to repeat the karyotypes and have the images of them.

Reviewer 3 Report

This manuscript is a careful descirption of a thorough study. I have only tow minor recommendations.

First the authors should check that the oval cover the eyes of the patient entirely.

Second in the legend to Figure 1d the authors should indicate what part of chromosome 16 is depicted.

Author Response

This manuscript is a careful description of a thorough study. I have only two minor recommendations.

First the authors should check that the oval cover the eyes of the patient entirely.

Second in the legend to Figure 1d the authors should indicate what part of chromosome 16 is depicted.

Response: We thank the reviewer for these valuable suggestions. We made the modifications as suggested: we changed the figure 1 in order to cover entirely the eyes of the patient and we added figure 3 to clarify with a scheme the region of chromosome 16 that is deleted.

Round 2

Reviewer 1 Report

No comments